# Knowledge of Oral Cancer Risk Factors among International Medical and Dental Students at the Lithuanian University of Health Sciences: A Cross-Sectional Study

**DOI:** 10.3390/healthcare11020271

**Published:** 2023-01-15

**Authors:** Sandra Petrauskienė, Dominika Kopczynska, Gintarė Žemgulytė, Kristina Saldūnaitė-Mikučionienė

**Affiliations:** 1Department of Preventive and Paediatric Dentistry, Faculty of Dentistry, Lithuanian University of Health Sciences, Lukšos-Daumanto 6, LT-50106 Kaunas, Lithuania; 2Department of Neurology, Medical Academy, Lithuanian University of Health Sciences, Eivenių 2, LT-50103 Kaunas, Lithuania

**Keywords:** oral cancer, knowledge, dental students, medical students

## Abstract

The aim of this study was to evaluate knowledge of oral cancer risk factors among international dental and medical students at the Lithuanian University of Health Sciences (LSMU). A cross-sectional study was conducted among international medical and dental students at the LSMU in 2018–2019. In total, 253 students participated, for a response rate of 76.2%. An anonymous self-administered English questionnaire, consisting of 18 items, covered background information (gender, faculty and academic study year), and statements about risk factors for oral cancer and attitudes towards the prevention and treatment of oral cancer. The fifteen statements had the following answer options: yes, no and I don’t know. The statistical data analysis was performed using SPSS version 22. Overall, a good, fair and poor knowledge of oral risk factors was reported by 15.02%, 56.92% and 28.06% of the participants, respectively. The mean score for knowledge of oral cancer risk factors was 10.01 (SD = 2.56), which was defined as fair knowledge. Cronbach’s alpha was found to be 0.78 (a good value). Bartlett’s sphericity test and the KMO index were adequate (χ^2^ = 677.563, *p* < 0.001; KMO = 0.788). This study demonstrates a lack of knowledge of oral cancer risk factors among international medical and dental students.

## 1. Introduction

Oral cancer is a complex and multifactorial disease. Oral cancer remains a challenging disease with low survival rates and a poor prognosis [1,2,3]. Worldwide, head and neck cancer is the 6th most common malignancy [4] and lip and oral cancer is the 16th most common malignancy [5]. Consequently, the incidence, mortality and disability-adjusted life years of oral cancer are steadily increasing worldwide [6].

Numerous risk factors, such as prolonged alcohol intake, tobacco exposure, diet, occupational activity, socioeconomic status, genetic susceptibility, poor oral hygiene and human papillomavirus (HPV) infection, can cause oral carcinogenesis [7]. Alcohol consumption and tobacco use have a (dose-associated) synergistic effect [8] that is related to the incidence of cancer mortality and morbidity [9]. It has been proven that even light alcohol intake increases the risk for oral cancer [8,10]. Furthermore, alcohol consumption and tobacco use increase the risk of cancer at various sites. Alcohol intake is defined as a risk factor for cancers in the oropharynx, larynx, esophagus, liver, colon, rectum and female breast. Long-term tobacco smoking tends to increase the risk for cancers of the lips, oral cavity, pharynx, larynx, lung, stomach, colon, rectum, breast, pancreas and liver [11]. Human papillomavirus (HPV) is a risk factor that contributes to the current increase in oropharyngeal cancer cases, especially among nonsmoking and nondrinking individuals [12]. Studies have revealed that oral health professionals have insufficient knowledge or low awareness of the link between HPV and oropharyngeal cancer [13,14].

Early-stage diagnosis is a key factor in reducing the incidence and in ensuring and increasing chances of survival [15]. Worldwide, cancer control strategies are launched by symptom awareness campaigns, primary prevention policies and extensive screening programs [16,17] However, these initiatives have an insignificant impact on improving cancer outcomes if early diagnosis interventions focus only on the symptoms of advanced-stage disease [17].

Despite easy access to the area for conventional oral examination, multiple factors contribute to the delay in oral cancer diagnosis, such as patient delay [18,19] and provider delay [18,20], which may be related to insufficient knowledge and inadequate awareness among health care providers [21]. The delay in cancer diagnosis can be identified as a risk factor for the mortality of oral cancer, due to postponed treatment delivery and the decreased survival rate of patients [22].

A more favorable prognosis for oral cancer treatment is associated with developed regions, due to a greater awareness among residents and more easily implemented multidisciplinary therapies for oral cancer patients [23,24]. Broadly speaking, prevention and early diagnosis are the most cost-effective long-term approaches to cancer prognosis [25].

Dental and medical students represent future health care professionals [26]. Thus, undergraduates in these fields should be assessed to determine deficient areas of oral cancer knowledge [26,27]. In addition, they should be encouraged to educate patients about potential risk factors for oral cancer, regarding its steadily increasing prevalence worldwide [6,26,27]. Previous studies have revealed that medical students tend to underestimate their role in patients’ education about smoking cessation and the prevention of various diseases [28]. Dental and medical school curricula should be enhanced and focus on raising oral cancer awareness among students, and training on oral cancer prevention and early detection should be provided. [29,30,31].

Subsequently, a number of international students and students from 87 countries, currently enrolled at Lithuanian University of Health Sciences, participated in this study [32]. This study aimed to evaluate the knowledge of oral cancer risk factors among international dental and medical students at the Lithuanian University of Health Sciences (LSMU).

## 2. Materials and Methods

A cross-sectional study was conducted to assess the knowledge of oral cancer risk factors among international medical and dental students at the Lithuanian University of Health Sciences during the 2018/2019 study year. The study was approved by the Bioethics Centre of the Lithuanian University of Health Sciences (No BEC-OF-50).

## 3. Subjects

The subjects were international dental and medical students at LSMU. The inclusion criteria for the subjects were dental and medical students studying in English and willing to participate.

A total of 166 international dental students and 690 international medical students studied at LSMU during the 2018/2019 academic year. An equal number of participants from both faculties (166 international dental students and 166 international medical students selected randomly) were enrolled in this study as a representative sample. The principal investigator (DK) asked all the participants (*n* = 332) to complete an anonymous self-administered written questionnaire before the beginning of a lecture or a compulsory practical class.

The aim of the study was explained to the participants before they completed the questionnaire. Participation was voluntary and anonymous; thus, the return of a completed questionnaire and the consent signed by subjects were considered as acceptance to participate. Overall, 253 completed questionnaires were returned.

Thus, 115 international dental students and 138 international medical students participated in this study. The response rate was 76.2%.

Regarding to the curriculum of odontology, the international dental students were dichotomized according to the year of study into “preclinical” (the 1st and 2nd years) and “clinical” (the 3rd, 4th and 5th years) phases [33]. International medical students were not divided into groups in relation to the peculiarities of the medicine programme [34].

## 4. Questionnaire

The questionnaire was developed by the authors (SP and DK). An anonymous self-administered English questionnaire, consisting of 18 items, covered background information (gender, faculty and academic study year), and statements about risk factors for oral cancer and attitudes towards the prevention and treatment of oral cancer.

Fifteen questions (statements) had the following answer options: yes, no and I don’t know.

The score for knowledge of oral cancer risk factors and clinical overview was calculated by giving a score of 1 point for each correctly chosen option and a score of 0 points for an incorrect option or an “I don’t know” response. The maximum score was 15.

Subsequently, the sum score was counted: scores < 9 were considered poor knowledge of oral cancer risk factors, scores from 9 to 12 were considered fair knowledge of oral cancer risk factors, and scores > 12 were considered good knowledge of oral cancer risk factors.

## 5. Statistical Analysis

Statistical data analysis was conducted using SPSS (Statistical Package for the Social Sciences for Windows, Chicago, IL, USA) version 22. To establish relationships between the categorical variables, chi-squared tests (χ^2^) were used. A *p* value ≤ 0.05 was set to indicate statistically significant differences. The Mann-Whitney U test was used to compare the mean scores between groups.

Cronbach’s alpha served as a measure of the internal consistency of the questionnaire and was found to be 0.78 (a good value). To calculate the sphericity index, Bartlett’s test was used with a significance level of *p* < 0.05. The Kaiser-Meyer-Olkin (KMO) measure was used to measure sample adequacy. In this study, both Bartlett’s sphericity test and the KMO index were adequate (χ^2^ = 677.563, *p* <0.001; KMO = 0.788) to perform the analysis.

Fifteen variables/items were included for score calculation. Answers to the items were given both positively and negatively.

For factor validity, items were segregated by predetermined categories: gender (male), certain types of HPV and oral-genital contacts that can lead to oral cancer (F1); environmental and habitual risk factors (hot spicy food, biting of cheek or lips and UV exposure increase the risk and dentures do not increase the risk of oral cancer) (F2); clinical overview of oral cancer (for example, early diagnosis improves recovery, oral cancer can be prevented and treated, necessity of additional training) (F3); and primary risk factors for oral cancer (tobacco usage and alcohol consumption), and oral cancer manifestation without initial complaints (F4). Factor validity was examined using a principal component analysis (PCA), using the varimax rotation with Kaiser normalization.

## 6. Results

Participants were distributed quite equally with regard to their academic year of study. International dental and medical students in the 2nd and 5th academic years of study prevailed among the participants (*p* > 0.05) (Table 1). Considering gender, females (53.91%) prevailed among international dental students, while males (54.35%) dominated among international medical students. Overall, the participants in this survey were 50.59% male and 49.41% female (*p* > 0.05) (Table 1).

A good, fair and poor knowledge of oral risk factors was reported by 15.02%, 56.92% and 28.06% of the participants, respectively.

The mean score for knowledge of oral cancer risk factors was 10.01 (SD = 2.56), defined as fair knowledge. Women showed an insignificantly better knowledge of oral cancer risk factors compared to men (10.20 (SD = 2.52) vs. 9.84 (SD = 2.58) (*p* = 0.338)) (Table 2). With regard to the faculty, this score was higher among international medical students than dental students (10.09 (SD = 2.62) vs. 9.92 (SD = 2.49) (*p* = 0.562). The clinical-year dental international students showed the best knowledge among all the participants (11.13 (SD = 1.85)), whereas preclinical-year dental students had the lowest score (8.23 (SD = 2.27)) (*p* < 0.0001) (Table 2).

A factorial analysis with four factors was preferred. As shown in Table 3, all the primary factor loadings were at least 0.462, and no items cross-loaded onto other factors. Factor loadings ranged between 0.462 and 0.779.

Factor 1 included sex (male), certain types of HPV, and oral-genital contact that can lead to oral cancer. The test–retest reliability among all participants was 0.623 for gender (male), 0.602 for certain types of HPV, 0.544 for oral–genital contact increasing the risk of oral cancer and 0.587 for the non-contagiousness of oral cancer.

Factor 2 was directly related to environmental and habitual risk factors for oral cancer. The test–retest reliability among all participants was 0.779 for hot spicy food, 0.717 for biting of the cheek or lips and 0.571 for UV exposure increasing the risk for oral cancer; the test–retest reliability was 0.462 for dentures not being a risk factor for oral cancer.

Factor 3 presented the clinical overview. The test–retest reliability among all participants was 0.746 for recovery improvement by early diagnosis, 0.697 for the need for additional training, 0.544 for possibilities to treat oral cancer and 0.497 for the preventability of oral cancer.

Factor 4 summarized the main risk factors (such as tobacco usage and alcohol consumption) and potential complaints of patients. The test–retest reliability among all participants was 0.669 for tobacco usage and 0.475 for alcohol consumption and 0.558 for oral cancer manifestation without initial complaints.

## 7. Discussion

This study revealed that the majority of international dental and medical students had a fair knowledge of oral cancer risk factors. Meanwhile, the score for knowledge of oral cancer risk was higher among international medical students than dental students. Senior dental students showed significantly better knowledge than younger students. This result was in line with other studies assessing awareness/knowledge of risk factors and the prevention of oral cancer [26,35,36,37].

Knowledge of oral cancer risk factors among dental and medical students varies worldwide. A study conducted by Sallam et al. revealed good knowledge of oral cancer among senior dental undergraduates in Jordan [38], while other studies showed insufficient awareness of risk factors for oral cancer among dental students in Malaysia, Turkey and Nepal [39,40,41].

The inadequate attention paid to oral cancer in the medical curriculum [29] may be associated with a lower level of knowledge among medical students than among dental students, particularly of risk factors, early diagnosis and oral changes related to oral cancer [42]. A study carried out in Saudi Arabia revealed an unsatisfactory knowledge of oral cancer and a lack of confidence in performing an oral examination among medical students [28]. However, this study showed that medical students had better self-reported knowledge than dental undergraduates.

Cancer prevention is based on the identification of and reduction in the contributing risk factors. Smoking was identified as a main risk factor for oral cancer by the vast majority of dental and medical students in this study, and this finding was in line with other studies [26,35,36,39,40,41,42]. Knowledge of alcohol as a risk factor varied from 14.5% [39] to 94% [26,35,36,40,41,42] among dental and medical students. The risk of alcohol consumption tends to be underestimated by medical students [42], and similar results were found in this study.

The findings of various studies have revealed that dental students show different attitudes towards HPV-related oral cancer. HPV was correctly identified as a risk factor for oral cancer by 40.3% [35,43] to 91.1% of dental students [26,36,44]. A study performed in Poland revealed that dental students did not relate the higher incidence of oral cancer to oral sex [43], while the findings of this study showed that international dental and medical students tended to agree about the association of oral–genital contact with an increased risk of oral cancer. The awareness of HPV may be raised with changes in the curriculum and training workshops [38].

A number of studies have revealed that only a minority of dental students worldwide identified immunosuppression, advanced age, sun exposure and poor diet as risk factors for oral cancer [26,35,39,40,41,45]. In this study, dental and medical students agreed that hot spicy food and frequent biting of the lips and cheeks may increase the risk for oral cancer, as in another study performed in Turkey [40]. Another study defined the same factors as non-risk factors for oral cancer [35]. In this study, poorly fitting dentures were identified as a non-risk factor by a minority of undergraduates, similar to the results of a study conducted in Romania [35].

The literature reveals a lack of confidence and inadequate knowledge among dental and medical undergraduates regarding oral cancer [27,39,46]. For instance, only a minority of dental students in Romania reported that oral cancer may manifest without initial complaints, as in this study [35]. Moreover, various studies have revealed that a majority of medical and dental undergraduates reported the necessity of additional training [27,42], similar to the results of our study. Adopting educational methods, such as smartphone applications, may be a key to engaging and improving diagnostic skills for oral cancer among undergraduates [27,39,47].

Unfortunately, the same pattern is evident among graduated practitioners. Although alcohol consumption, tobacco use, older age, viral infection and prior oral cancer were identified as risk factors for oral cancer by the majority of dentists [45,48], they had an inadequate knowledge of diagnostic abilities and early detection practices [30,49,50].

A need for updated teaching methodologies remains crucial for several reasons, such as low current insight on oral cancer among the majority of general medical and dental practitioners and low interest in continuing education courses on oral cancer after graduation [20,46,51]. A higher level of knowledge about oral cancer would ensure appropriate oral cancer screening for patients in the future [36].

In addition, patients who receive advice about oral cancer from dentists are likely to accept it as a principal source of information [52]. Community engagement in prevention programs may also increase oral cancer awareness in the population [53]. Nevertheless, translating knowledge of oral cancer into effective diagnostic ability is still a challenge [46]. Insufficient knowledge of oral cancer among undergraduates may be a critical issue worldwide [40].

This study was the first to assess and evaluate self-reported knowledge of oral cancer risk factors among international dental and medical students at the Lithuanian University of Health Sciences. The findings of this survey may be used as baseline data for future research. This study enrolled international dental and medical undergraduates at LSMU and covered all study years with a response rate of 76.2%, which can be considered high.

However, this study has certain limitations. The questionnaire contained closed-ended questions, which may limit and narrow the relevance of the responses. All data were self-reported and subjective. Therefore, the generalizability of our study may be limited.

## 8. Conclusions

International dental and medical students had a fair self-reported knowledge of oral cancer risk factors. Medical students were found to have a higher knowledge of oral cancer risk factors than dental ones. Knowledge of oral cancer risk factors among dental students was enhanced with an increase in the study year. Furthermore, both medical and dental undergraduates reported the necessity of additional training and showed interest in improving their current knowledge of oral cancer risk factors. This study highlights the need to raise awareness of oral cancer risk factors among dental and medical undergraduates.

## Figures and Tables

**Table 1 healthcare-11-00271-t001:** Characteristics of the participants by gender and academic year.

Variables	Faculty				Total	
	International Dental Students (N = 115)		International Medical Students (N = 138)			
	N	%	N	%	N	%
Gender						
Male	53	46.09	75	54.35	128	50.59
Female	62	53.91	63	45.65	125	49.41
Total	115	100	138	100	253	100
Academic year of study						
1	23	20	22	15.94	45	17.79
2	25	21.74	24	17.39	49	19.37
3	21	18.26	26	18.84	47	18.58
4	21	18.26	27	19.57	48	18.97
5	25	21.74	29	21.01	54	21.34
6	0	0	10	7.25	10	3.95
Total	115	100	138	100	253	100

*p* value—comparison between participants (*p* > 0.05).

**Table 2 healthcare-11-00271-t002:** The score for knowledge of oral cancer risk factors among participants at LSMU (N = 253).

Variables	MS (SD)	*p* Value
Gender		
Male	9.84 (2.58)	0.338
Female	10.20 (2.52)	
Faculty		
Odontology	9.92 (2.49)	0.562
Medicine	10.09 (2.62)	
International dental students		
Preclinical years	8.23 (2.27)	<0.0001
Clinical years	11.13 (1.85)	

MS—mean score (brackets contain standard deviation); statistical analysis by Mann-Whitney U test.

**Table 3 healthcare-11-00271-t003:** Factorial analysis of the questionnaire.

Item/Statement	Component			
	F1	F2	F3	F4
Gender (male) is related to a higher risk of oral cancer	**0.623**	−0.099	0.162	0.024
Certain types of HPV can lead to oral cancer	**0.602**	0.224	0.084	0.317
Oral cancer is noncontagious	**0.587**	0.185	0.009	−0.325
Oral–genital contact increases the risk of oral cancer	**0.544**	0.352	0.026	0.129
Hot spicy food increases the risk of oral cancer	−0.103	**0.779**	0.172	−0.054
Biting of cheek or lips increases the risk of oral cancer	0.137	**0.717**	0.118	0.121
UV exposure increases the risk of oral cancer	0.393	**0.571**	−0.005	0.182
Dentures are not a risk for oral cancer	0.400	**0.462**	0.148	−0.122
Early diagnosis improves recovery	−0.129	0.109	**0.746**	0.038
Additional training is necessary	0.001	0.045	**0.697**	−0.237
Oral cancer can be treated	0.202	0.107	**0.544**	0.226
Oral cancer can be prevented	0.381	0.025	**0.497**	0.027
Tobacco usage is a risk factor for oral cancer	−0.134	−0.056	−0.060	**0.669**
Alcohol consumption is a risk factor for oral cancer	0.370	0.377	−0.022	**0.475**
Oral cancer manifests without initial complaints	0.216	0.162	0.325	**0.558**

HPV—Human papillomavirus (a possible contributor to this phenomenon); UV—Ultraviolet; Note: Factor loadings over 40 appear in bold. Extraction method: principal component analysis. Rotation method: varimax with Kaiser normalization. KMO measure of sampling adequacy = 0.788. Bartlett’s test of sphericity (χ^2^) = 677.563, *p* < 0.001.

## Data Availability

The datasets used and/or analyzed during the current study are available from the corresponding author on reasonable request.

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
