# Peer review of "Knowledge of Oral Cancer Risk Factors among International Medical and Dental Students at the Lithuanian University of Health Sciences: A Cross-Sectional Study"

_healthcare, 2023, doi:10.3390/healthcare11020271_

Round 1
Reviewer 1 Report
The paper of Petrauskienė et al. is to assess the knowledge about oral cancer among dental and medical students in Lithuania. Overall, the paper is well-written and scientifically sound. It is unknown why the authors only selected international students and excluded domestic students. Further comments that may help the authors improve their paper are in the attached file.

Author Response
Our responses to the Reviewers’ comments
We thank the reviewers for their constructive comments. Please find enclosed our responses to the reviewers highlighted in red.
Page: 1
Number: 1 Being future health care professional is not relevant to the context, therefore, a sentence about the significant of knowledge and early detection of oral cancer for better patient management is required. Considering the maximum of total number of words (200) and longer description of questionnaire this sentence was excluded in the manuscript. (Page 1)
Number: 2 this is just a question in my mind and no further action is required from the authors. Why only included international students not domestic students? knowledge is knowledge regardless of the studied language, especially because oral caner is a global health burden
Number: 3 brief description for the questionnaire is required. how many questions? open or closed questions?- description of questionnaire was included as recommended in the abstract. (Page 1).
Number: 4 6th most common what? It is amended- 6th most common malignancy. (Page 1).
Page: 2
Number: 1 This paragraph is a bit vague. Reasons behind these requirements are necessary. For example, why undergraduates should be encouraged to educate patients about oral cancers? Are there studies showed that they do not do? Do current curricula lack focus onto oral cancer and therefore should be revised ? Patients‘ education about prevention of oral cancer is essential, therefore dental and medical students they should be encouraged to educate patients about potential risk factors for oral cancer. Study performed by Kujan et al. [28] revealed that majority medical students believed it to be beyond their role to aid patients in smoking cessation measures or to take part in other disease preventative strategies. Therefore, dental and medical school curricula should be enhanced and focus on raising oral cancer awareness among students and training on oral cancer prevention and early detection.
This paragraph was revised as recommended. (Page 2).
Number: 2 please attach the questionnaire- The questionnaire will be attached as supplement.
Page: 3
Number: 1 please do not start the sentence by numbers: The sentence is amended as recommended. (Page 3).
Page: 4
Number: 1 please add "standard deviation" or "SD" where applicable: SD was added where applicable. (Pages 1 and 4).
Page: 6
Number: 1 as the country "Jordan" was mentioned, please mention the countries for studies 34-36, OR remove the countries for all studies- The missing countries were mentioned as recommended in the manuscript. (Page 6).
Number: 2 previous studies showed overall unsatisfactorily level of knowledge about oral cancer among medical students. DOI: 10.1007/s13187-013-0527-4 Please include these studies and discuss their results- the study was included and the main findings were presented as recommended. (Page 7).
Page: 7
Number: 1 again, what is the point of including only international, not domestic students?
The number of international students at LSMU is increasing. Students from 87 countries are currently enrolled. Some Lithuanian students graduated gymnasium in Lithuania or secondary education in foreign country decide and study Odontology or Medicine curriculum in English at Lithuanian University of Health Sciences.
Previous studies (by Zaborskis et al., Mustakallio et al.) showed the differences of attitude about addictiveness of smoking and pecularities of dental treatment between native and international students due to the multicultural approach and background. Furthermore, study carried out by Petrauskiene et al. [37] revealed relatevely low awareness about oral carcer among international dental and medical students at LSMU. Therefore we aimed to evaluate the knowledge of oral cancer risk factors among international dental and medical students at LSMU. (Page 2).
Zaborskis A, Volkyte A, Narbutaite J, Virtanen JI. Smoking and attitudes towards its cessation among native and international dental students in Lithuania. BMC Oral Health. 2017 Jul 11;17(1):106. doi: 10.1186/s12903-017-0397-y. PMID: 28693469; PMCID: PMC5504839.
Mustakallio S, Näpänkangas R, Narbutaite J, Virtanen JI. Graduating dentists' perceptions about their professional competence in Finland and Lithuania. Eur J Dent Educ. 2020 May;24(2):227-232. doi: 10.1111/eje.12488. Epub 2020 Jan 13. PMID: 31845488.
Number: why? please include justification for this finding in the discussion- results of this study showed, that score of knowledge of oral cancer risk was higher among international medical students than dental ones. The justification was included in the discussion as recommended. (Page 6).
Our responses to the Reviewers’ comments
We thank the reviewers for their constructive comments. Please find enclosed our responses to the reviewers highlighted in red.
Page: 1
Number: 1 Being future health care professional is not relevant to the context, therefore, a sentence about the significant of knowledge and early detection of oral cancer for better patient management is required. Considering the maximum of total number of words (200) and longer description of questionnaire this sentence was excluded in the manuscript. (Page 1)
Number: 2 this is just a question in my mind and no further action is required from the authors. Why only included international students not domestic students? knowledge is knowledge regardless of the studied language, especially because oral caner is a global health burden
Number: 3 brief description for the questionnaire is required. how many questions? open or closed questions?- description of questionnaire was included as recommended in the abstract. (Page 1).
Number: 4 6th most common what? It is amended- 6th most common malignancy. (Page 1).
Page: 2
Number: 1 This paragraph is a bit vague. Reasons behind these requirements are necessary. For example, why undergraduates should be encouraged to educate patients about oral cancers? Are there studies showed that they do not do? Do current curricula lack focus onto oral cancer and therefore should be revised ? Patients‘ education about prevention of oral cancer is essential, therefore dental and medical students they should be encouraged to educate patients about potential risk factors for oral cancer. Study performed by Kujan et al. [28] revealed that majority medical students believed it to be beyond their role to aid patients in smoking cessation measures or to take part in other disease preventative strategies. Therefore, dental and medical school curricula should be enhanced and focus on raising oral cancer awareness among students and training on oral cancer prevention and early detection.
This paragraph was revised as recommended. (Page 2).
Number: 2 please attach the questionnaire- The questionnaire will be attached as supplement.
Page: 3
Number: 1 please do not start the sentence by numbers: The sentence is amended as recommended. (Page 3).
Page: 4
Number: 1 please add "standard deviation" or "SD" where applicable: SD was added where applicable. (Pages 1 and 4).
Page: 6
Number: 1 as the country "Jordan" was mentioned, please mention the countries for studies 34-36, OR remove the countries for all studies- The missing countries were mentioned as recommended in the manuscript. (Page 6).
Number: 2 previous studies showed overall unsatisfactorily level of knowledge about oral cancer among medical students. DOI: 10.1007/s13187-013-0527-4 Please include these studies and discuss their results- the study was included and the main findings were presented as recommended. (Page 7).
Page: 7
Number: 1 again, what is the point of including only international, not domestic students?
The number of international students at LSMU is increasing. Students from 87 countries are currently enrolled. Some Lithuanian students graduated gymnasium in Lithuania or secondary education in foreign country decide and study Odontology or Medicine curriculum in English at Lithuanian University of Health Sciences.
Previous studies (by Zaborskis et al., Mustakallio et al.) showed the differences of attitude about addictiveness of smoking and pecularities of dental treatment between native and international students due to the multicultural approach and background. Furthermore, study carried out by Petrauskiene et al. [37] revealed relatevely low awareness about oral carcer among international dental and medical students at LSMU. Therefore we aimed to evaluate the knowledge of oral cancer risk factors among international dental and medical students at LSMU. (Page 2).
Zaborskis A, Volkyte A, Narbutaite J, Virtanen JI. Smoking and attitudes towards its cessation among native and international dental students in Lithuania. BMC Oral Health. 2017 Jul 11;17(1):106. doi: 10.1186/s12903-017-0397-y. PMID: 28693469; PMCID: PMC5504839.
Mustakallio S, Näpänkangas R, Narbutaite J, Virtanen JI. Graduating dentists' perceptions about their professional competence in Finland and Lithuania. Eur J Dent Educ. 2020 May;24(2):227-232. doi: 10.1111/eje.12488. Epub 2020 Jan 13. PMID: 31845488.
Number: why? please include justification for this finding in the discussion- results of this study showed, that score of knowledge of oral cancer risk was higher among international medical students than dental ones. The justification was included in the discussion as recommended. (Page 6).
Our responses to the Reviewers’ comments
We thank the reviewers for their constructive comments. Please find enclosed our responses to the reviewers highlighted in red.
Page: 1
Number: 1 Being future health care professional is not relevant to the context, therefore, a sentence about the significant of knowledge and early detection of oral cancer for better patient management is required. Considering the maximum of total number of words (200) and longer description of questionnaire this sentence was excluded in the manuscript. (Page 1)
Number: 2 this is just a question in my mind and no further action is required from the authors. Why only included international students not domestic students? knowledge is knowledge regardless of the studied language, especially because oral caner is a global health burden
Number: 3 brief description for the questionnaire is required. how many questions? open or closed questions?- description of questionnaire was included as recommended in the abstract. (Page 1).
Number: 4 6th most common what? It is amended- 6th most common malignancy. (Page 1).
Page: 2
Number: 1 This paragraph is a bit vague. Reasons behind these requirements are necessary. For example, why undergraduates should be encouraged to educate patients about oral cancers? Are there studies showed that they do not do? Do current curricula lack focus onto oral cancer and therefore should be revised ? Patients‘ education about prevention of oral cancer is essential, therefore dental and medical students they should be encouraged to educate patients about potential risk factors for oral cancer. Study performed by Kujan et al. [28] revealed that majority medical students believed it to be beyond their role to aid patients in smoking cessation measures or to take part in other disease preventative strategies. Therefore, dental and medical school curricula should be enhanced and focus on raising oral cancer awareness among students and training on oral cancer prevention and early detection.
This paragraph was revised as recommended. (Page 2).
Number: 2 please attach the questionnaire- The questionnaire will be attached as supplement.
Page: 3
Number: 1 please do not start the sentence by numbers: The sentence is amended as recommended. (Page 3).
Page: 4
Number: 1 please add "standard deviation" or "SD" where applicable: SD was added where applicable. (Pages 1 and 4).
Page: 6
Number: 1 as the country "Jordan" was mentioned, please mention the countries for studies 34-36, OR remove the countries for all studies- The missing countries were mentioned as recommended in the manuscript. (Page 6).
Number: 2 previous studies showed overall unsatisfactorily level of knowledge about oral cancer among medical students. DOI: 10.1007/s13187-013-0527-4 Please include these studies and discuss their results- the study was included and the main findings were presented as recommended. (Page 7).
Page: 7
Number: 1 again, what is the point of including only international, not domestic students?
The number of international students at LSMU is increasing. Students from 87 countries are currently enrolled. Some Lithuanian students graduated gymnasium in Lithuania or secondary education in foreign country decide and study Odontology or Medicine curriculum in English at Lithuanian University of Health Sciences.
Previous studies (by Zaborskis et al., Mustakallio et al.) showed the differences of attitude about addictiveness of smoking and pecularities of dental treatment between native and international students due to the multicultural approach and background. Furthermore, study carried out by Petrauskiene et al. [37] revealed relatevely low awareness about oral carcer among international dental and medical students at LSMU. Therefore we aimed to evaluate the knowledge of oral cancer risk factors among international dental and medical students at LSMU. (Page 2).
Zaborskis A, Volkyte A, Narbutaite J, Virtanen JI. Smoking and attitudes towards its cessation among native and international dental students in Lithuania. BMC Oral Health. 2017 Jul 11;17(1):106. doi: 10.1186/s12903-017-0397-y. PMID: 28693469; PMCID: PMC5504839.
Mustakallio S, Näpänkangas R, Narbutaite J, Virtanen JI. Graduating dentists' perceptions about their professional competence in Finland and Lithuania. Eur J Dent Educ. 2020 May;24(2):227-232. doi: 10.1111/eje.12488. Epub 2020 Jan 13. PMID: 31845488.
Number: why? please include justification for this finding in the discussion- results of this study showed, that score of knowledge of oral cancer risk was higher among international medical students than dental ones. The justification was included in the discussion as recommended. (Page 6).
Our responses to the Reviewers’ comments
We thank the reviewers for their constructive comments. Please find enclosed our responses to the reviewers highlighted in red.
Page: 1
Number: 1 Being future health care professional is not relevant to the context, therefore, a sentence about the significant of knowledge and early detection of oral cancer for better patient management is required. Considering the maximum of total number of words (200) and longer description of questionnaire this sentence was excluded in the manuscript. (Page 1)
Number: 2 this is just a question in my mind and no further action is required from the authors. Why only included international students not domestic students? knowledge is knowledge regardless of the studied language, especially because oral caner is a global health burden
Number: 3 brief description for the questionnaire is required. how many questions? open or closed questions?- description of questionnaire was included as recommended in the abstract. (Page 1).
Number: 4 6th most common what? It is amended- 6th most common malignancy. (Page 1).
Page: 2
Number: 1 This paragraph is a bit vague. Reasons behind these requirements are necessary. For example, why undergraduates should be encouraged to educate patients about oral cancers? Are there studies showed that they do not do? Do current curricula lack focus onto oral cancer and therefore should be revised ? Patients‘ education about prevention of oral cancer is essential, therefore dental and medical students they should be encouraged to educate patients about potential risk factors for oral cancer. Study performed by Kujan et al. [28] revealed that majority medical students believed it to be beyond their role to aid patients in smoking cessation measures or to take part in other disease preventative strategies. Therefore, dental and medical school curricula should be enhanced and focus on raising oral cancer awareness among students and training on oral cancer prevention and early detection.
This paragraph was revised as recommended. (Page 2).
Number: 2 please attach the questionnaire- The questionnaire will be attached as supplement.
Page: 3
Number: 1 please do not start the sentence by numbers: The sentence is amended as recommended. (Page 3).
Page: 4
Number: 1 please add "standard deviation" or "SD" where applicable: SD was added where applicable. (Pages 1 and 4).
Page: 6
Number: 1 as the country "Jordan" was mentioned, please mention the countries for studies 34-36, OR remove the countries for all studies- The missing countries were mentioned as recommended in the manuscript. (Page 6).
Number: 2 previous studies showed overall unsatisfactorily level of knowledge about oral cancer among medical students. DOI: 10.1007/s13187-013-0527-4 Please include these studies and discuss their results- the study was included and the main findings were presented as recommended. (Page 7).
Page: 7
Number: 1 again, what is the point of including only international, not domestic students?
The number of international students at LSMU is increasing. Students from 87 countries are currently enrolled. Some Lithuanian students graduated gymnasium in Lithuania or secondary education in foreign country decide and study Odontology or Medicine curriculum in English at Lithuanian University of Health Sciences.
Previous studies (by Zaborskis et al., Mustakallio et al.) showed the differences of attitude about addictiveness of smoking and pecularities of dental treatment between native and international students due to the multicultural approach and background. Furthermore, study carried out by Petrauskiene et al. [37] revealed relatevely low awareness about oral carcer among international dental and medical students at LSMU. Therefore we aimed to evaluate the knowledge of oral cancer risk factors among international dental and medical students at LSMU. (Page 2).
Zaborskis A, Volkyte A, Narbutaite J, Virtanen JI. Smoking and attitudes towards its cessation among native and international dental students in Lithuania. BMC Oral Health. 2017 Jul 11;17(1):106. doi: 10.1186/s12903-017-0397-y. PMID: 28693469; PMCID: PMC5504839.
Mustakallio S, Näpänkangas R, Narbutaite J, Virtanen JI. Graduating dentists' perceptions about their professional competence in Finland and Lithuania. Eur J Dent Educ. 2020 May;24(2):227-232. doi: 10.1111/eje.12488. Epub 2020 Jan 13. PMID: 31845488.
Number: why? please include justification for this finding in the discussion- results of this study showed, that score of knowledge of oral cancer risk was higher among international medical students than dental ones. The justification was included in the discussion as recommended. (Page 6).
Our responses to the Reviewers’ comments
We thank the reviewers for their constructive comments. Please find enclosed our responses to the reviewers highlighted in red.
Page: 1
Number: 1 Being future health care professional is not relevant to the context, therefore, a sentence about the significant of knowledge and early detection of oral cancer for better patient management is required. Considering the maximum of total number of words (200) and longer description of questionnaire this sentence was excluded in the manuscript. (Page 1)
Number: 2 this is just a question in my mind and no further action is required from the authors. Why only included international students not domestic students? knowledge is knowledge regardless of the studied language, especially because oral caner is a global health burden
Number: 3 brief description for the questionnaire is required. how many questions? open or closed questions?- description of questionnaire was included as recommended in the abstract. (Page 1).
Number: 4 6th most common what? It is amended- 6th most common malignancy. (Page 1).
Page: 2
Number: 1 This paragraph is a bit vague. Reasons behind these requirements are necessary. For example, why undergraduates should be encouraged to educate patients about oral cancers? Are there studies showed that they do not do? Do current curricula lack focus onto oral cancer and therefore should be revised ? Patients‘ education about prevention of oral cancer is essential, therefore dental and medical students they should be encouraged to educate patients about potential risk factors for oral cancer. Study performed by Kujan et al. [28] revealed that majority medical students believed it to be beyond their role to aid patients in smoking cessation measures or to take part in other disease preventative strategies. Therefore, dental and medical school curricula should be enhanced and focus on raising oral cancer awareness among students and training on oral cancer prevention and early detection.
This paragraph was revised as recommended. (Page 2).
Number: 2 please attach the questionnaire- The questionnaire will be attached as supplement.
Page: 3
Number: 1 please do not start the sentence by numbers: The sentence is amended as recommended. (Page 3).
Page: 4
Number: 1 please add "standard deviation" or "SD" where applicable: SD was added where applicable. (Pages 1 and 4).
Page: 6
Number: 1 as the country "Jordan" was mentioned, please mention the countries for studies 34-36, OR remove the countries for all studies- The missing countries were mentioned as recommended in the manuscript. (Page 6).
Number: 2 previous studies showed overall unsatisfactorily level of knowledge about oral cancer among medical students. DOI: 10.1007/s13187-013-0527-4 Please include these studies and discuss their results- the study was included and the main findings were presented as recommended. (Page 7).
Page: 7
Number: 1 again, what is the point of including only international, not domestic students?
The number of international students at LSMU is increasing. Students from 87 countries are currently enrolled. Some Lithuanian students graduated gymnasium in Lithuania or secondary education in foreign country decide and study Odontology or Medicine curriculum in English at Lithuanian University of Health Sciences.
Previous studies (by Zaborskis et al., Mustakallio et al.) showed the differences of attitude about addictiveness of smoking and pecularities of dental treatment between native and international students due to the multicultural approach and background. Furthermore, study carried out by Petrauskiene et al. [37] revealed relatevely low awareness about oral carcer among international dental and medical students at LSMU. Therefore we aimed to evaluate the knowledge of oral cancer risk factors among international dental and medical students at LSMU. (Page 2).
Zaborskis A, Volkyte A, Narbutaite J, Virtanen JI. Smoking and attitudes towards its cessation among native and international dental students in Lithuania. BMC Oral Health. 2017 Jul 11;17(1):106. doi: 10.1186/s12903-017-0397-y. PMID: 28693469; PMCID: PMC5504839.
Mustakallio S, Näpänkangas R, Narbutaite J, Virtanen JI. Graduating dentists' perceptions about their professional competence in Finland and Lithuania. Eur J Dent Educ. 2020 May;24(2):227-232. doi: 10.1111/eje.12488. Epub 2020 Jan 13. PMID: 31845488.
Number: why? please include justification for this finding in the discussion- results of this study showed, that score of knowledge of oral cancer risk was higher among international medical students than dental ones. The justification was included in the discussion as recommended. (Page 6).
Our responses to the Reviewers’ comments
We thank the reviewers for their constructive comments. Please find enclosed our responses to the reviewers highlighted in red.
Page: 1
Number: 1 Being future health care professional is not relevant to the context, therefore, a sentence about the significant of knowledge and early detection of oral cancer for better patient management is required. Considering the maximum of total number of words (200) and longer description of questionnaire this sentence was excluded in the manuscript. (Page 1)
Number: 2 this is just a question in my mind and no further action is required from the authors. Why only included international students not domestic students? knowledge is knowledge regardless of the studied language, especially because oral caner is a global health burden
Number: 3 brief description for the questionnaire is required. how many questions? open or closed questions?- description of questionnaire was included as recommended in the abstract. (Page 1).
Number: 4 6th most common what? It is amended- 6th most common malignancy. (Page 1).
Page: 2
Number: 1 This paragraph is a bit vague. Reasons behind these requirements are necessary. For example, why undergraduates should be encouraged to educate patients about oral cancers? Are there studies showed that they do not do? Do current curricula lack focus onto oral cancer and therefore should be revised ? Patients‘ education about prevention of oral cancer is essential, therefore dental and medical students they should be encouraged to educate patients about potential risk factors for oral cancer. Study performed by Kujan et al. [28] revealed that majority medical students believed it to be beyond their role to aid patients in smoking cessation measures or to take part in other disease preventative strategies. Therefore, dental and medical school curricula should be enhanced and focus on raising oral cancer awareness among students and training on oral cancer prevention and early detection.
This paragraph was revised as recommended. (Page 2).
Number: 2 please attach the questionnaire- The questionnaire will be attached as supplement.
Page: 3
Number: 1 please do not start the sentence by numbers: The sentence is amended as recommended. (Page 3).
Page: 4
Number: 1 please add "standard deviation" or "SD" where applicable: SD was added where applicable. (Pages 1 and 4).
Page: 6
Number: 1 as the country "Jordan" was mentioned, please mention the countries for studies 34-36, OR remove the countries for all studies- The missing countries were mentioned as recommended in the manuscript. (Page 6).
Number: 2 previous studies showed overall unsatisfactorily level of knowledge about oral cancer among medical students. DOI: 10.1007/s13187-013-0527-4 Please include these studies and discuss their results- the study was included and the main findings were presented as recommended. (Page 7).
Page: 7
Number: 1 again, what is the point of including only international, not domestic students?
The number of international students at LSMU is increasing. Students from 87 countries are currently enrolled. Some Lithuanian students graduated gymnasium in Lithuania or secondary education in foreign country decide and study Odontology or Medicine curriculum in English at Lithuanian University of Health Sciences.
Previous studies (by Zaborskis et al., Mustakallio et al.) showed the differences of attitude about addictiveness of smoking and pecularities of dental treatment between native and international students due to the multicultural approach and background. Furthermore, study carried out by Petrauskiene et al. [37] revealed relatevely low awareness about oral carcer among international dental and medical students at LSMU. Therefore we aimed to evaluate the knowledge of oral cancer risk factors among international dental and medical students at LSMU. (Page 2).
Zaborskis A, Volkyte A, Narbutaite J, Virtanen JI. Smoking and attitudes towards its cessation among native and international dental students in Lithuania. BMC Oral Health. 2017 Jul 11;17(1):106. doi: 10.1186/s12903-017-0397-y. PMID: 28693469; PMCID: PMC5504839.
Mustakallio S, Näpänkangas R, Narbutaite J, Virtanen JI. Graduating dentists' perceptions about their professional competence in Finland and Lithuania. Eur J Dent Educ. 2020 May;24(2):227-232. doi: 10.1111/eje.12488. Epub 2020 Jan 13. PMID: 31845488.
Number: why? please include justification for this finding in the discussion- results of this study showed, that score of knowledge of oral cancer risk was higher among international medical students than dental ones. The justification was included in the discussion as recommended. (Page 6).

Reviewer 2 Report
Thank you for the interesting paper. Please correct on p. 3-145 the insignificance, the values look indeed significantly different?!
References should be presented in a selected style, please avoid different text formats (in capitals vs. non-capitals). Is the reference 46 indeed in English?
There are two critical comments: Why the authors excluded all lithuanian students from the same faculty, it should be discussed at least.
And why the medical students were not divided in preclinical vs. clinical groups, again, should be discussed.
Author Response
Our responses to the Reviewers’ comments
We thank the reviewers for their constructive comments. Please find enclosed our responses to the reviewers highlighted in red.
Reviewers' Comments to Author:
( ) I would not like to sign my review report
(x) I would like to sign my review report
English language and style
( ) English very difficult to understand/incomprehensible
( ) Extensive editing of English language and style required
( ) Moderate English changes required
(x) English language and style are fine/minor spell check required
( ) I don't feel qualified to judge about the English language and style
Yes |
Can be improved |
Must be improved |
Not applicable |
|
Does the introduction provide sufficient background and include all relevant references? |
(x) |
( ) |
( ) |
( ) |
Are all the cited references relevant to the research? |
(x) |
( ) |
( ) |
( ) |
Is the research design appropriate? |
(x) |
( ) |
( ) |
( ) |
Are the methods adequately described? |
(x) |
( ) |
( ) |
( ) |
Are the results clearly presented? |
( ) |
(x) |
( ) |
( ) |
Are the conclusions supported by the results? |
(x) |
( ) |
( ) |
( ) |
Comments and Suggestions for Authors
Please correct on p. 3-145 the insignificance, the values look indeed significantly different?! P-value was added as inaccuracy, because we did not aimed to compare the values (prevalence of poor, fair and good knowledge among all participants). (Page 4).
References should be presented in a selected style, please avoid different text formats (in capitals vs. non-capitals). Is the reference 46 indeed in English?- References were corrected as recommended ( in non-capital letters).
Reference 46 [amended number 51] is written in English: https://sbdmj.lsmuni.lt/182/182-03.pdf
There are two critical comments: Why the authors excluded all lithuanian students from the same faculty, it should be discussed at least.
The number of international students at LSMU is increasing. Students from 87 countries are currently enrolled. Previous studies (by Zaborskis et al., Mustakallio et al.) showed the differences of attitude about addictiveness of smoking and pecularities of dental treatment between native and international students due to the multicultural approach and background. Furthermore, study carried out by Petrauskiene et al. [37] revealed relatively low awareness about oral carcer among international dental and medical students at LSMU. Therefore we aimed to evaluate the knowledge of oral cancer risk factors among international dental and medical students at LSMU .
Zaborskis A, Volkyte A, Narbutaite J, Virtanen JI. Smoking and attitudes towards its cessation among native and international dental students in Lithuania. BMC Oral Health. 2017 Jul 11;17(1):106. doi: 10.1186/s12903-017-0397-y. PMID: 28693469; PMCID: PMC5504839.
Mustakallio S, Näpänkangas R, Narbutaite J, Virtanen JI. Graduating dentists' perceptions about their professional competence in Finland and Lithuania. Eur J Dent Educ. 2020 May;24(2):227-232. doi: 10.1111/eje.12488. Epub 2020 Jan 13. PMID: 31845488.
And why the medical students were not divided in preclinical vs. clinical groups, again, should be discussed. Dental students were divided into preclinical and clinical groups regarding to the curriculum of odontology. The preclinical stage consists of basic and biomedical studies and lasts for the first two years. During the three-year clinical stage, students study not only the specialty subjects such as prevention of oral diseases, paediatric dentistry, orthodontics, oral and maxillofacial surgery, prosthodontics, cariology, endodontics, periodontology and diseases of oral mucosa and receive practical training, both in the phantom-head laboratory and in the clinics, but also internal diseases, ear nose and throat pathology and skin diseases.
Meanwhile, the curriculum of medicine is comprised of basic sciences during the first year; from the second year, the studies are problem-based, integrating basic and clinical subjects and the analysis of real clinical situations. Knowledge is applied and skills are developed during the fourth to sixth years in solving clinical problems. In addition, only the sixth-year medical undergraduates study the subject related to oral health in clinic of oral and maxillofacial surgery. Therefore, medical students were not divided into preclinical and clinical groups. (Page 3).

Reviewer 3 Report
Thank you for inviting me to review this manuscript.
My recommendations are:
- Does the term international means that only students from other countries were included in the study? Is there a reason why only international students were included in the study? Please explain, and if there is a specific reason, it should be described in the paper.
- The description of the exclusion criteria should remove the sentence "students of other faculties," given that the inclusion criteria established that only dental and medical students were considered for the research.
- Are local students excluded from this study?
- The dental students were divided into two groups meanwhile the medical students were not. The authors made no explanation for this decision.
- Which method was followed to select the 166 medical students to be included in the study? Please describe.
- Was there a methodology to establish the outcomes from the questionnaire (poor, fair, good)? Is there a similar study using this kind of methodology? Please explain.
My primary concern is that the questionary used to establish knowledge levels has yet to be validated. Without validation, the poor, fair, and good outcomes are just arbitrary decisions, and it is impossible to confirm the level of knowledge from it. Therefore I suggest rejecting this paper.
Author Response
Our responses to the Reviewers’ comments
We thank the reviewers for their constructive comments. Please find enclosed our responses to the reviewers highlighted in red.
Reviewers' Comments to Author:
(x) I would not like to sign my review report
( ) I would like to sign my review report
English language and style
( ) English very difficult to understand/incomprehensible
( ) Extensive editing of English language and style required
( ) Moderate English changes required
( ) English language and style are fine/minor spell check required
(x) I don't feel qualified to judge about the English language and style
Yes |
Can be improved |
Must be improved |
Not applicable |
|
Does the introduction provide sufficient background and include all relevant references? |
(x) |
( ) |
( ) |
( ) |
Are all the cited references relevant to the research? |
(x) |
( ) |
( ) |
( ) |
Is the research design appropriate? |
( ) |
( ) |
( ) |
(x) |
Are the methods adequately described? |
( ) |
( ) |
(x) |
( ) |
Are the results clearly presented? |
( ) |
( ) |
(x) |
( ) |
Are the conclusions supported by the results? |
( ) |
( ) |
( ) |
(x) |
Comments and Suggestions for Authors
.
My recommendations are:
- Does the term international means that only students from other countries were included in the study? Is there a reason why only international students were included in the study? Please explain, and if there is a specific reason, it should be described in the paper.
- The description of the exclusion criteria should remove the sentence "students of other faculties," given that the inclusion criteria established that only dental and medical students were considered for the research.
- Are local students excluded from this study?
- The dental students were divided into two groups meanwhile the medical students were not. The authors made no explanation for this decision.
- Which method was followed to select the 166 medical students to be included in the study? Please describe.
- Was there a methodology to establish the outcomes from the questionnaire (poor, fair, good)? Is there a similar study using this kind of methodology? Please explain.
My primary concern is that the questionary used to establish knowledge levels has yet to be validated. Without validation, the poor, fair, and good outcomes are just arbitrary decisions, and it is impossible to confirm the level of knowledge from it. Therefore I suggest rejecting this paper.
- Does the term international means that only students from other countries were included in the study? Is there a reason why only international students were included in the study? Please explain, and if there is a specific reason, it should be described in the paper.
The term international students means that medical or dental students study in English. Some Lithuanian students graduated gymnasium in Lithuania or secondary education in foreign country decide and study Odontology or Medicine curriculum in English at Lithuanian University of Health Sciences.
Subsequently, the number of international students at LSMU is increasing. Students from 87 countries are currently enrolled. Previous studies (by Zaborskis et al., Mustakallio et al.) showed the differences of attitude about addictiveness of smoking and pecularities of dental treatment between native and international students due to the multicultural approach and background. Furthermore, study carried out by Petrauskiene et al. [37] revealed relatively low awareness about oral carcer among international dental and medical students at LSMU. Therefore we aimed to evaluate the knowledge of oral cancer risk factors among international dental and medical students at LSMU. (Page 2)
Zaborskis A, Volkyte A, Narbutaite J, Virtanen JI. Smoking and attitudes towards its cessation among native and international dental students in Lithuania. BMC Oral Health. 2017 Jul 11;17(1):106. doi: 10.1186/s12903-017-0397-y. PMID: 28693469; PMCID: PMC5504839.
Mustakallio S, Näpänkangas R, Narbutaite J, Virtanen JI. Graduating dentists' perceptions about their professional competence in Finland and Lithuania. Eur J Dent Educ. 2020 May;24(2):227-232. doi: 10.1111/eje.12488. Epub 2020 Jan 13. PMID: 31845488.
- The description of the exclusion criteria should remove the sentence "students of other faculties," given that the inclusion criteria established that only dental and medical students were considered for the research- The exclusion criteria „Students of other faculties“ was removed as recommended. (Page 2)
- Are local students excluded from this study? Students studying Odontology and Medicine in Lithuanian were excluded.
- The dental students were divided into two groups meanwhile the medical students were not. The authors made no explanation for this decision. Dental students were divided into preclinical and clinical groups regarding to the curriculum of odontology. The preclinical stage consists of basic and biomedical studies and lasts for the first two years. During the three-year clinical stage, students study not only the specialty subjects such as prevention of oral diseases, paediatric dentistry, orthodontics, oral and maxillofacial surgery, prosthodontics, cariology, endodontics, periodontology and diseases of oral mucosa and receive practical training, both in the phantom-head laboratory and in the clinics, but also internal diseases, ear nose and throat pathology and skin diseases.
Meanwhile, the curriculum of medicine is comprised of basic sciences during the first year; from the second year, the studies are problem-based, integrating basic and clinical subjects and the analysis of real clinical situations. Knowledge is applied and skills are developed during the fourth to sixth years in solving clinical problems. In addition, only the sixth-year medical undergraduates study the subject related to oral health in clinic of oral and maxillofacial surgery. Therefore, medical students were not divided into preclinical and clinical groups. (Page 3).
- Which method was followed to select the 166 medical students to be included in the study? Please describe. 166 medical students were selected randomly from 690 medical students (each fourth student according to students list) and according their willingness to participate. (Page 3).
- Was there a methodology to establish the outcomes from the questionnaire (poor, fair, good)? Is there a similar study using this kind of methodology? Please explain. References [26] and [37] had used similar kind of methodology.
Studies performed by Shadid et al and by Shubayr had used 60% as cut-off points to assess the knowledge level. They described good knowledge when sum score was less than 60% and poor knowledge when sum score was more than 60%. The same principle was used in this study.
Shadid RM, Abu Ali MA, Kujan O. Knowledge, attitudes, and practices of oral cancer prevention among dental students and interns: an online cross‑sectional questionnaire in Palestine. BMC Oral Health. 2022 Sep 5;22(1):381. doi: 10.1186/s12903-022-02415-8. PMID: 36064693; PMCID: PMC9446528.
Shubayr MA, Bokhari AM, Essa AA, Nammazi AM, Al Agili DE. Knowledge, attitudes, and practices of oral cancer prevention among students, interns, and faculty members at the college of dentistry of Jazan University. BMC Oral Health. 2021 Dec 1;21(1):612. doi: 10.1186/s12903-021-01973-7. PMID: 34852821; PMCID: PMC8638461.

Round 2
Reviewer 3 Report
I have no more suggestions.